# A Well-Defined H9N2 Avian Influenza Virus Genotype with High Adaption in Mammals was Prevalent in Chinese Poultry Between 2016 to 2019

**DOI:** 10.3390/v12040432

**Published:** 2020-04-11

**Authors:** Zhaokun Chen, Qinghua Huang, Shaohua Yang, Shuai Su, Baoquan Li, Ning Cui, Chuantian Xu

**Affiliations:** 1Shandong Key Laboratory of Animal Disease Control & Breeding, Institute of Animal Science and Veterinary Medicine, Shandong Academy of Agricultural Sciences, Jinan 250100, China; dczk521@126.com (Z.C.); huangqih@163.com (Q.H.); ysh7865@163.com (S.Y.); 2College of Life Sciences, Shandong Normal University, Jinan 250014, China; 3Shandong Provincial Key Laboratory of Animal Biotechnology and Disease Control and Prevention, College of Veterinary Medicine, Shandong Agricultural University, Tai’an 271018, China; ssu6307@163.com (S.S.); libq72@163.com (B.L.)

**Keywords:** H9N2 subtype avian influenza virus, pathogenicity, virus titer, mammal, genotype

## Abstract

H9N2 subtype avian influenza virus (AIV) is widely prevalent in poultry, and the virus is becoming adaptive to mammals, which poses pandemic importance. Here, BALB/c mice were employed as a model to evaluate the adaption in mammals of 21 field H9N2 viruses isolated from avian species between 2016 to 2019 in China. The replication capacity of the viruses was evaluated in the lungs of mice. The pathogenicity of the viruses were compared by weight loss and lung lesions from infected mice. The whole genomic sequences of the viruses were further characterized to define the associated phenotypes of the H9N2 viruses in vitro and in vivo. The results showed that most viruses could replicate well and cause lesions in the mouse lungs. The propagation capacity in MDCK cells and damage to respiratory tissues of the infected mice corresponded to relative viral titers in the mouse lungs. Further genome analysis showed that all of the H9N2 viruses belonged to the same genotype, G57, and contained a couple of amino acid substitutions or deletions that have been demonstrated as avian-human markers. Additionally, nine amino acids residues in seven viral proteins were found to be correlated with the replication phenotypes of the H9N2 viruses in mammals. The study demonstrated that a well-defined H9N2 AIV genotype with high adaption in mammals was prevalent in China in recent years. Further investigations on the role of the identified residues and continuous surveillance of newly identified mutations associated with host adaption should be strengthened to prevent any devastating human influenza pandemics.

## 1. Introduction

H9N2 influenza viruses were first isolated from turkeys in North America in 1966 [1]. Infections of H9N2 avian influenza virus (AIV) strains in domestic poultry in Mainland China became established during the mid-1990s, and have become the most prevalent subtype of influenza viruses in multiple avian species [2,3,4]. Although H9N2 viruses are classified as low-pathogenicity AIV, they sometimes cause severe disease in poultry, occasionally accompanied by a marked reduction in egg production and high mortality [5,6]. Importantly, viruses of endemic H9N2 sublineages circulating in poultry populations in Asia occasionally transmit to humans and mammals [6,7]. H9N2 subtype viruses were isolated from pigs in 1998 [8] and, subsequently, were isolated from humans with influenza-like illness in Hong Kong and Mainland China [9,10]. The possession of human-like receptor specificity and occasional transmission of H9N2 AIV to mammalian species raise public concerns on the potential threat of this virus in humans [7,11,12,13,14]. In addition, H9N2 AIVs are considered as significant donors contributing to the emergence of a novel subtype of AIV that has human health implications [15]. Sequence analysis showed that H9N2 viruses may provide the internal genes to the human H5N1, H7N9, and H10N8 viruses [15,16,17]. Therefore, the persistence of the H9N2 virus has a significant impact on poultry industries and public health.

In the present study, 21 representative strains of H9N2 influenza viruses isolated from diseased chickens or ducks in different farms in Shandong and the surrounding area during 2016 to 2019 were selected for investigation of their pathogenicity in mice with the aim to investigate the adaption and the potential pathogenicity of the novel H9N2 reassortants to mammals.

## 2. Materials and Methods

### 2.1. Ethics Statement

All animal experiments were carried out in strict accordance with the recommendations in the Guide for the Care and Use of Laboratory Animals of the Ministry of Science and Technology of the People’s Republic of China. The protocol used in the study was approved by the Review Board of Shandong Veterinary Research Institute, Shandong Academy of Agricultural Sciences.

### 2.2. Viruses, Cells, and Animals

H9N2 subtype AIV strains were isolated from clinical samples, including liver and kidney tissues taken from farms across Shandong province and peripheral regions of China between 2016 to 2019 (Table 1). Madin-Darby canine kidney (MDCK) cells were preserved in our lab. Ten-day-old embryonated Specific-pathogen-free (SPF) embryos were used for virus isolation, purification, and proliferation. BALB/c mice were bred at the Experimental Animal Center of Shandong Veterinary Research Institute.

### 2.3. Experimental Design

A total of 21 isolates were tested for the adaption of prevalent H9N2 AIV strains in mammalian hosts. Viral replication capacities were determined in 10-day-old embryonated chicken eggs and Madin-Darby canine kidney (MDCK) cells. BALB/c mice were employed as a model to evaluate the replication capacity and potential pathogenicity in the mammalian hosts of the prevalent H9N2 AIV isolates. The viral genome and mutations of the isolates were further analyzed to explore the genotypes correlated with the phenotypes of these H9N2 viruses in vitro and in vivo of mammals.

### 2.4. 50% Egg Infectious Dose (EID_50_) Determination

The allantoic fluid in virus-infected embryos was tested by hemagglutination assay as described previously [18]. Positive allantoic fluid was harvested and determined for EID_50_ in 10-day-old embryonated chicken eggs. Briefly, the allantoic fluid was diluted by phosphate buffer saline (PBS) at continuous gradient dilution from 10^−1^ to 10^−10^. Each dilution of the allantoic fluid was inoculated into five embryos and incubated for 72 h at 37 °C. Virus infection was tested by haemagglutination assay and EID_50_ was then calculated by the method of Reed and Muench [19].

### 2.5. 50% Tissue Infectious Dose (TCID_50_) Determination

The allantoic fluid containing 10^6^ EID_50_ of each virus was diluted by phosphate buffer saline (PBS) at continuous gradient dilution from 10^−1^ to 10^−10^. Each dilution of the allantoic fluid was inoculated into four replicates of MDCK cell monolayer in the presence of 4 µg/mL tosylsulfonyl phenylalanyl chloromethyl ketone-treated trypsin (Sigma, Santa clara, USA). Relative virus replication was determined by measuring the HA titer of the supernatant after 48 h incubation at 37 °C with 5% CO_2_. TCID_50_ was calculated by the method of Reed and Muench [19].

### 2.6. Sample Collection 

To investigate the adaption and potential pathogenicity of 21 prevalent H9N2 AIV strains to mammals, 242 six-week-old female BALB/c mice were randomly divided into 22 groups, with 11 in each group. 10^6^ EID_50_ of each virus in a volume of 50 µL was intranasally inoculated into the exposure groups. Mice inoculated with equal volumes of PBS were used as negative controls. Lungs from three mice of each group were collected under sterile conditions at three and six days post inoculation (dpi) for virus titration and pathology study, and the remaining five mice were monitored daily for weight loss for two weeks. The experiment was repeated at least twice.

### 2.7. Determination of Viral Titers in Tissues of Mice

Viral titers in tissues of infected mice were determined for EID_50_ in 10-day-old embryonated chicken eggs as described previously [3]. Briefly, the organs collected were homogenized in 1 mL of cold phosphate-buffered saline (PBS). Solid debris was pelleted by centrifugation at 4000× *g* for 5 min, and the supernatant were used for virus titration in embryos. Virus titers were given in units of log_10_EID_50_ per 1 mL ± standard deviation (SD).

### 2.8. Pathology Study 

Lung samples were fixed in 10% neutral buffer formalin, and paraffin tissue sections were made. Sections were stained with hematoxylin and eosin (HE) for examination under a microscope (Nikon, Tokyo, Japan) with a NIKON digital sight DS-FI2 imaging system.

### 2.9. Genotype and Mutation Definition

Viral genotypes were analyzed for the H9N2 viruses using gene phylogenic analysis and determined according to a previous study [20]. Briefly, based on the branching posterior and reported clade reference viruses, phylogenetic analyses of the eight viral segments were respectively conducted using the Clustal alignment algorithm and the neighbor-joining method (MEGA version 6, Arizona State University, Tempe, USA). Estimates of the phylogenic relationships were calculated by the neighbor-joining bootstrap method with 1000 replicates. Phylogenetic trees were automatically partitioned into clusters using a 20th percentile cutoff. Genotype identification numbers were assigned according to the previous study [20]. Viral genotypes of the prevalent H9N2 viruses were determined by the combination of clade assignments of each of the eight segments. The deduced amino acid sequences from eight segments were aligned and analyzed using the Lasergene software (DNASTAR, Madison, WI, USA).

### 2.10. Statistical Analysis 

All data are presented as mean per group ± standard deviation. An analysis of variance was performed to determine significant differences (*p* < 0.05) between groups by GraphPad Prism software (Prism version 6. GraphPad Software, San Diego, CA, USA, www.graphpad.com.). One-way ANOVA with the Tukey’s multiple comparison test was used for statistical comparisons of virus titers and gene transcription levels. 

## 3. Results

### 3.1. Determination of Viral Replication Capacities

Investigation of the viral replication capacities suggested that the prevalent H9N2 viruses grew comparably in eggs, while viral titers in MDCK cells varied from 10^1^ to 10^9^ TCID_50_/0.2 mL (Figure 1). Sixteen isolates grew to ≥ 10^4^ TCID_50_ in MDCK cells. The CK/SD/3426/16 virus did not replicate well in MDCK cells with the viral titers lower than 10^1^ TCID_50_/0.2 mL. 

### 3.2. Viral Titer in the Lungs of Mice

The replication capacities of these H9N2 isolates in mice were evaluated by the virus titer in the lungs of each infected mice group. Lung virus titers were determined for three mice each at 3 and 6 dpi for each group as summarized in Figure 2. Of the 21 H9N2 isolates, only two viruses, CK/SD/3426/16 and CK/SD/1642/18, could not be recovered from the mouse lungs at any time point. The remaining 19 viruses showed varied replication capacities in the lungs of mice, with more than 50% of isolates replicating well in the lungs (titers > 3.0 log_10_EID_50_). The CK/SD/3424/16 could replicate in the mouse lungs with the highest titer of 5.78 log_10_EID_50_ at 3 dpi.

### 3.3. Body Weight of Mice

The influence of infection with the prevalent H9N2 strains on the body weights of mice was determined and is summarized in Figure 3. All mice infected with isolated H9N2 AIV strains survived infection. However, most of the infected mice showed obvious weight loss during the observation period, and the CK/SD/3424/16 caused the greatest weight loss (more than 10%) from 4 to 8 dpi among these strains. The body weight reached the same level as the mock-infected control groups at 13 dpi.

### 3.4. Pathology of Infected Mice

The pathogenicity of the prevalent H9N2 viruses in mice was further demonstrated by H&E staining of lung tissues collected at 3 dpi (Figure 4). The morphology and structure of bronchioles in most of the virus-infected lung sections were normal, except in the CK/SX/1757/19-infected ones, in which a few necrotic cell fragments were seen in the bronchioles and inflammatory cell infiltration was evident around the blood vessels and bronchioles, forming inflammatory cuffs. A small amount of inflammatory cell infiltration can be seen in 15% to 90% of the area in the alveolus tissue from most infected mice, among which the percentage of CK/SD/3424/16, DC/SD/915/17, and CK/SX/1757/19 reached more than 80%. Slight thickening of the alveolar walls was observed for the CK/SD/1286/16, DC/SD/1432/17, and CK/SX/1757/19-infected mice, and moderate thickening of the alveolar walls was observed for the CK/SD/3424/16, DC/SD/915/17, and CK/SD/1647/18-infected mice. In addition, A/Duck/Shandong/915/2017 and CK/SD/1646/18-infected lungs exhibited blood capillary congestion. Alveolar stenosis and blood capillary congestion was seen in CK/SX/1757/19-infected mice. Collectively, the most severe pathology was observed for the CK/SD/3424/16, DC/SD/915/17, and CK/SX/1757/19-infected groups.

### 3.5. Genotype

To clarify the reason for the varied replication capability of these H9N2 isolates in vitro and in vivo in mammals, the viral genomes of the isolates were analyzed. It is worth noting that the eight segments of all H9N2 viruses were derived from the same original viruses (Figure 5). HA segments of the viruses were derived from CK/ST/1579/00, CK/JS/1/00, and CK/GD/10/00. NA segments were derived from CK/BJ/1/94 and CK/HK/739/94. In terms of the internal genes, the PB2 fragments were derived from DC/ST/7488/04 and DC/ST/163/04, and the M fragments were derived from Quail/HK/G1/97 virus. CK/HN/nd/98, CK/SH/F/98, and CK/SJZ/2/98 provided the remaining four internal genes for all of the prevalent isolates in our study. Collectively, all isolates were classified into the G57 reference genotype according to the defined criterion, and there are no links between the genotypes and the phenotypes of these H9N2 viruses.

We further investigated the avian–human signature positions that contribute to cross barriers between species (Table 2). All these H9N2 viruses had the avian-like residues 627E and 701D in the PB2 protein. However, all viruses had a uniform three-amino-acid deletion at position 62 to 64 in the NA stalk region and the same amino acids of 89V, 309D, 477G in the PB2 protein, 13P in the PB1 protein, 224S, 351E, 356R in the PA protein, 165N, 226L, 316S in the HA protein (H3 numbering), 30D, 215A in the M1 protein, and 149A in the NS1 protein. Most isolates harbored 292V and 588V residues in the PB2 protein, but no strong correlation with the replication phenotypes in mammals.

On the basis of the replication capability of these H9N2 isolates in MDCK cells and in mice, detailed analysis and comparison of deduced amino acid sequences from eight segments of H9N2 AIV were carried out. Nine additional amino acids of the entire genome that may be correlated with the replication phenotypes in mammals were identified and are listed in Figure 6. The PB2 I676M and V726I, PA A70V, HA L87P, D239N, and I422V, NP V217I and V239M, and NA I189V mutants may serve as associated residues for the increased replication of H9N2 viruses in mammals.

## 4. Discussion

H9N2 subtype AIV is widely prevalent in poultry, and the virus is becoming adaptive to mammals [7,9,10]. We used BALB/c mice as a model to evaluate the replication capacity in mammalian hosts of the field H9N2 viruses isolated from avian species in China. The viral genomes of the isolates were further analyzed to clarify the associated biological phenotypes in mammals.

The genotypic diversity of AIVs fluctuated owing to the frequent reassortment characteristics of the viruses [20,21]. The BJ/94-like and G1-like viruses were the dominant genotypes circulating in poultry throughout China between the mid-1990s to the mid-2000s [13,22]. Previous studies reported that G9 and G1 viruses were highly pathogenic to mice since they were able to replicate to high titers in the mouse lungs directly without prior adaption [22,23]. Further analysis of the viral genome showed that these G9- and G1-like H9N2 AIVs possessed internal genes similar to the highly pathogenic H5N1 virus from Hong Kong in 1997 [3,22]. However, the diversified genotypes of the F/98 series were evolutionarily selected and become predominant in China after 2006, while the genotypes of the other series gradually decreased [20]. The G57 genotype from the F/98 series was first found in 2007, and the majority of the recently circulated H9N2 AIVs in China were classed into genotype G57 viruses [24]. Consistently, all of the H9N2 isolates between 2016 to 2019 in our study belonged to genotype G57 viruses, further indicating that the G57 genotype harbors high genetic compatibility to be a dominant strain. However, genetically similar H9N2 AIVs frequently showed different pathogenicity in mice [25,26,27]. Teng reported that only three H9N2 isolates among the 25 viruses tested between 2009 to 2012 in the genotype B69, which shared the same source of gene fragments as genotype G57, replicated efficiently in mice [27]. However, most viruses isolated between 2016 to 2019 in our study could replicate well in the lungs of mice. The results of propagation capacity in MDCK cells and damage to respiratory tissues of the infected mice corresponded to relative viral titers in the mouse lungs. The results of our present study combined with previous reports indicated that the genotype G57 H9N2 viruses have shown changed biological properties and have gradually acquired higher adaption in mammals [27]. It can be concluded that after entering a specific country, H9N2 viruses give rise to well-defined genetic groups that optimally infect poultry, and continue to evolve to adapt to wide range of host populations. Continuous surveillance of the transboundary infectious capacity of prevalent H9N2 virus should be strengthened to prevent any devastating human influenza pandemics like the 2009 swine-origin H1N1 IAV pandemic [28].

Reassortment of gene segments between several influenza viruses in conjunction with multiple adaptive mutations in the virus genome are considered the typical way for an influenza virus to cross species barriers, adapt to new hosts, and potentially increase virulence [29,30,31]. Previous studies have demonstrated that H9N2 viruses with different segment reassortments formed diverse genotypes and presented different pathogenicity [25,32]. Although all of the H9N2 viruses in our study possessed high gene homology and belonged to the same genotype, they still showed different biological characteristics. Some of the viruses replicated slowly both in vitro and in vivo in mammalian tissue and some could not even be recovered from the mouse lungs, while other viruses replicated well in the mouse lungs with severe pathogenic lesion and weight loss. Single substitution or a combination of several amino acid changes have been shown to increase replication or virulence of AIV in new hosts [14,33,34,35]. We then analyzed the coding proteins from all of the eight AIV fragments in our H9N2 field isolates for a couple of amino acid substitutions that have been demonstrated as avian–human markers (Table 2) [14,25,33,34,36,37,38,39,40,41,42,43,44]. All of the H9N2 viruses had the avian-like residues 627E and 701D in the PB2 protein, which are key factors for mammalian adaption [35]. However, all these H9N2 viruses contained a three-amino-acid deletion at position 62 to 64 in the NA stalk region and possessed at least 13 amino acid residues in seven viral proteins (PB2, PB1, PA, HA, NA, M1, NS1) that contribute to increased polymerase activity, change the receptor binding specificity, expand the cell tropism, or enhance the replication of influenza viruses in ma mmals. A shorter stalk NA was proved to improve NA enzyme activity and increase HA cleavage efficiency when combined with the 316S residue in HA1 [34]. It is worth noting that nine additional amino acid residues in seven viral proteins (PB2, PA, HA, NP, NA) were found to be correlated with the replication phenotypes of the H9N2 viruses in MDCK cells and in mice in the present study. Particularly, the 726I residue in the PB2 protein might provide the basis for the enhanced replication capacity as all of the viruses that replicated well in mice contained this residue. The PB2 protein plays a vital role in the host range and virulence of influenza viruses [45]. AIV can enhance polymerase activity in mammals by acquiring mutations in the viral polymerase complex of the PB1, PB2, and PA proteins [35]. The H9N2 viruses contained the 726I residue in PB2 protein and a combination of at least other three of the above mutated sites showed high viral titers in mammalian cells. The precise roles of these mutations remain to be determined. Plasmid-based reverse genetics and site-directed mutagenesis will be employed to further investigate their contributions to the dramatically altered virulence of prevalent H9N2 viruses. Newly identified mutations associated with host adaption should be considered highly important in case of a H9N2 AIV strain that is gradually generated by optimal combinations of mutations and evolved to jump between species.

In conclusion, BALB/c mice were employed as a model to evaluate the adaption of the field H9N2 viruses isolated from avian species between 2016 to 2019 in China to mammalian hosts. Most viruses could replicate well in the lungs of mice. The results of propagation capacity in MDCK cells and damage to respiratory tissues of the infected mice corresponded to relative viral titers in the mouse lungs. Further genomic analysis showed that all of the H9N2 viruses belonged to the same genotype, G57, and contained a couple of amino acid substitutions or deletions that have been demonstrated as avian–human markers. Additionally, nine amino acid residues in seven viral proteins were found to be correlated with the replication phenotypes of the H9N2 viruses in vitro and in vivo. Further investigation of the role of the identified residues and continuous surveillance of newly identified mutations that are associated with host adaption should be performed.

## Figures and Tables

**Figure 1 viruses-12-00432-f001:**
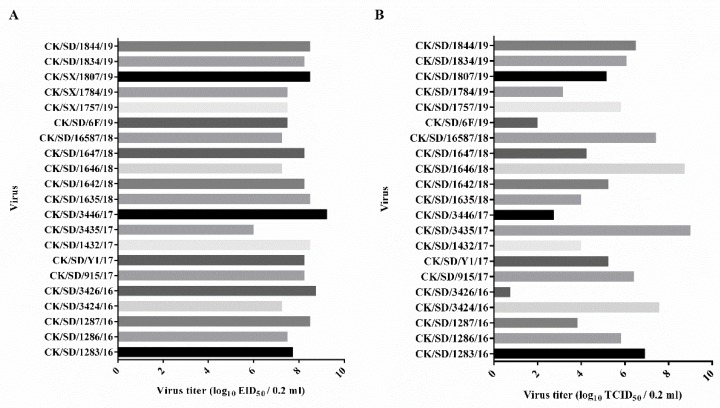
Viral replication capacities in embryonated chicken eggs and MDCK cells. Allantoic fluid containing the H9N2 viruses were determined for EID_50_ in 10-day-old embryonated chicken eggs (**A**) and TCID_50_ in MDCK cells (**B**). Virus titers were given in units of log_10_EID_50_ or log_10_TCID_50_ per 0.2 mL.

**Figure 2 viruses-12-00432-f002:**
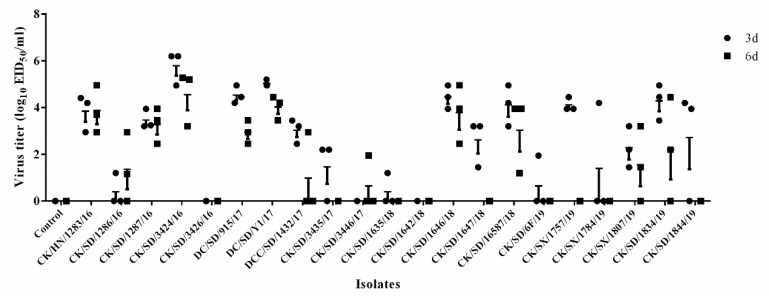
Viral titers in lungs of infected mice. Mice were intranasally inoculated with 10^6^ EID_50_ of each virus in a 50 µL volume. Three mice from each group were euthanized at 3 and 6 dpi. Titers of virus recovered from the supernatants of homogenized lung tissues are shown. Virus titers were given in units of log_10_EID_50_ per 1 mL ± standard deviation (SD).

**Figure 3 viruses-12-00432-f003:**
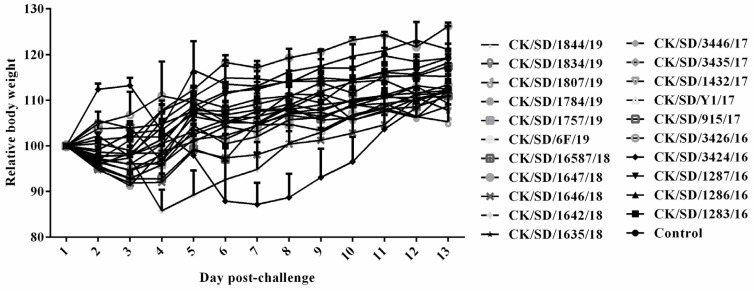
Body weight changes of mice. The values represent the average body weights compared to the baseline weight and standard deviations from five mice. Relative weight loss. Mice were intranasally inoculated with 10^6^ EID_50_ of virus or diluent (mock). The body weights of five inoculated mice were measured daily and are represented as the percentage of the weight on the day of inoculation (day 0). Six-week old female BALB/c mice were used in all experiments and were observed for 14 days post infection.

**Figure 4 viruses-12-00432-f004:**
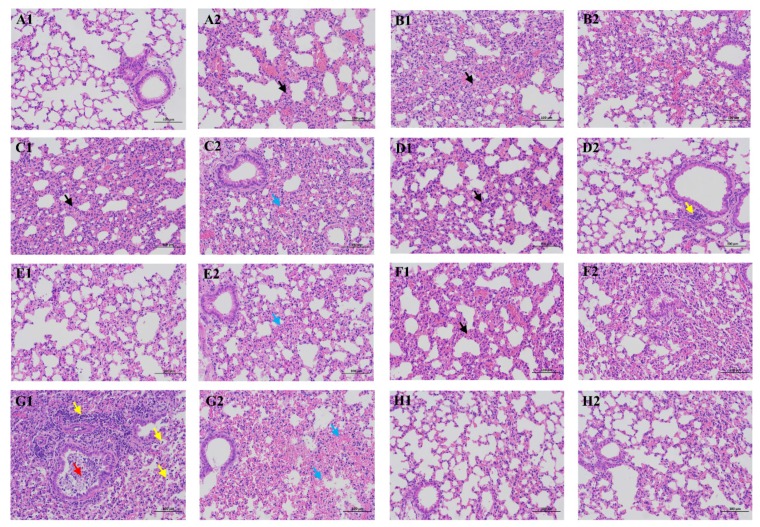
Representative histopathological changes in H&E-stained lung sections from mice infected with the indicated viruses at 3 dpi. Scale bar = 100 μm. Black arrows indicate alveolar walls with mild or moderate thickening. Blue arrows indicate alveolar walls with mild or moderate congestion and even bleeding. Yellow arrows indicate inflammatory cells that exhibit a small amount of effusion or point infiltration and even form inflammatory cuffs. Red arrows indicate necrotic cell debris. A-H: A1 indicates the control group. A2 and B to H indicate lung sections of mice infected with CK/SD/1286/16, CK/SD/3424/16, DC/SD/915/17, DC/SD/1432/17, CK/SD/1646/18, CK/SD/1647/18, CK/SX/1757/19, or CK/SD/1834/19.

**Figure 5 viruses-12-00432-f005:**
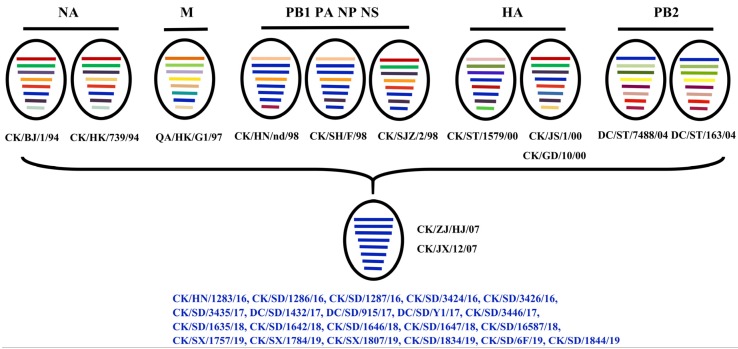
Genotypes of Chinese H9N2 viruses isolated from 2016 to 2019. Viral genotypes were analyzed for the H9N2 viruses based on gene phylogenic analysis and determined by the combination of clade assignments of each of the eight segments according to a previous study [20]. The eight gene segments are (horizontal bars from the top) PB1, PB2, PA, HA, NP, NA, M, and NS. Each color represents a viral segment clade.

**Figure 6 viruses-12-00432-f006:**
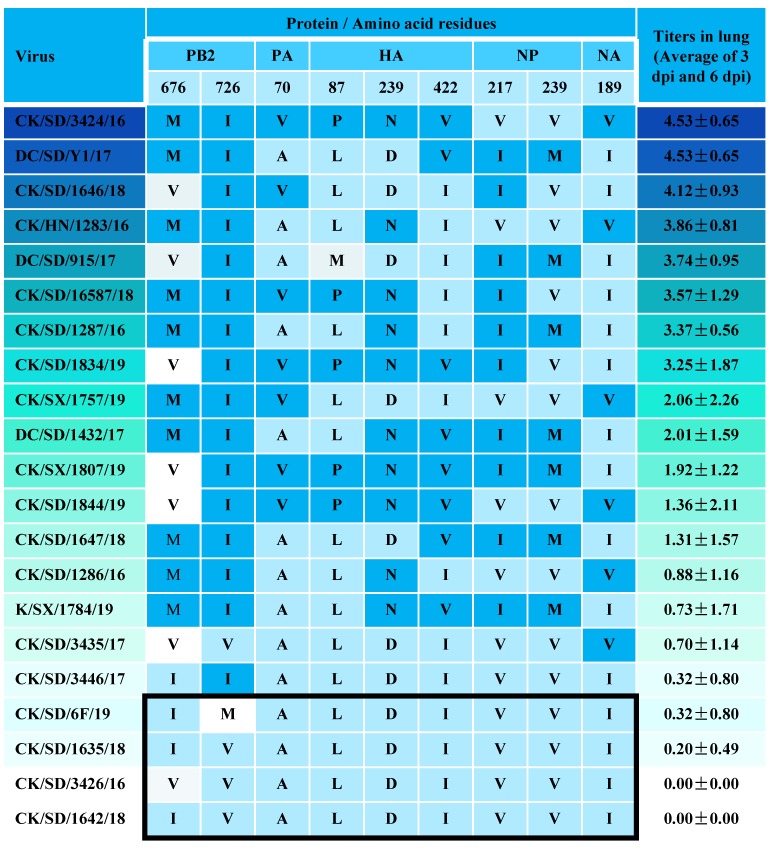
Amino acid residues of the H9N2 viruses that correlate with mouse replication phenotype. On the basis of viral titers at 3 dpi and 6 dpi of these H9N2 isolates in mouse lung, detailed analysis and comparison of deduced amino acid sequences from eight segments of H9N2 AIV were carried out. Dark blue represents viruses that replicate well in mice. Light blue represents viruses that replicate slowly in mice. Capital letters stand for amino acid abbreviations. M: Methionine, V: Valine, I: Isoleucine, A: Isoleucine, P: Proline, L: Leucine, N: Asparagine, D: Asparticacid.

**Table 1 viruses-12-00432-t001:** H9N2 avian influenza virus (AIV) strains used in this study.

Isolate	Abbreviation	Host	Date of Isolation	Genbank Accession No.
A/Chicken/Henan/1283/2016	CK/HN/1283/16	Chicken	February 2016	MH375347-MH375354
A/Chicken/Shandong/1286/2016	CK/SD/1286/16	Chicken	April 2016	MH375359-MH375366
A/Chicken/Shandong/1287/2016	CK/SD/1287/16	Chicken	April 2016	MH375368-MH375375
A/Chicken/Shandong/3424/2016	CK/SD/3424/16	Chicken	June 2016	MH667569-MH667576
A/Chicken/Shandong/3426/2016	CK/SD/3426/16	Chicken	July 2016	MH667582-MH667589
A/Chicken/Shandong/3435/2017	CK/SD/3435/17	Chicken	January 2017	MH667590-MH667597
A/Duck/Shandong/1432/2017	DC/SD/1432/17	Duck	April 2017	MH375444-MH375451
A/Duck/Shandong/915/2017	DC/SD/915/17	Duck	May 2017	MH375378-MH375385
A/Duck/Shandong/Y1/2017	DC/SD/Y1/17	Duck	May 2017	MH375416-MH375423
A/Chicken/Shandong/3446/2017	CK/SD/3446/17	Chicken	May 2017	MH889082-MH889089
A/Chicken/Shandong/1635/2018	CK/SD/1635/18	Chicken	April 2018	MK367676-MK367683
A/Chicken/Shandong/1642/2018	CK/SD/1642/18	Chicken	April 2018	MK367685-MK367692
A/Chicken/Shandong/1646/2018	CK/SD/1646/18	Chicken	May 2018	MK367659-MK367666
A/Chicken/Shandong/1647/2018	CK/SD/1647/18	Chicken	May 2018	MK367667-MK367674
A/Chicken/Shandong/16587/2018	CK/SD/16587/18	Chicken	May 2018	MK367643-MK367650
A/Chicken/ShanXi/1757/2019	CK/SX/1757/19	Chicken	January 2019	MN780494-MN780501
A/Chicken/ShanXi/1784/2019	CK/SX/1784/19	Chicken	March 2019	MN780586-MN780593
A/Chicken/ShanXi/1807/2019	CK/SX/1807/19	Chicken	April 2019	MN780832-MN780839
A/Chicken/Shandong/1834/2019	CK/SD/1834/19	Chicken	May 2019	MN765112-MN765119
A/Chicken/Shandong/6F/2019	CK/SD/6F/19	Chicken	May 2019	MN759632-MN759639
A/Chicken/Shandong/1844/2019	CK/SD/1844/19	Chicken	June 2019	MN765144-MN765151

**Table 2 viruses-12-00432-t002:** Analysis of key residues located at the avian–human signature positions in the H9N2 AIV isolates.

Protein	Site	H9N2 AIV Strain																				References
		3424	915	1283	1646	16587	1834	1287	1757	1432	1844	1807	1647	1286	3435	1784	6F	3446	1635	3426	1642	
PB2	89V	V	V	V	V	V	V	V	V	V	V	V	V	V	V	V	V	V	V	V	V	[33]
	292V	V	V	I	V	V	V	V	V	V	V	V	V	V	V	V	I	V	I	V	V	[36]
	309D	D	D	D	D	D	D	D	D	D	D	D	D	D	D	D	D	D	D	D	D	[33]
	477G	G	G	G	G	G	G	G	G	G	G	G	G	G	G	G	G	G	G	G	G	[33]
	588V	V	V	V	V	V	V	V	V	A	V	V	A	V	V	A	V	V	V	V	A	[37]
PB1	13P	P	P	P	P	P	P	P	P	P	P	P	P	P	P	P	P	P	P	P	P	[38]
PA	224S	S	S	S	S	S	S	S	S	S	S	S	S	S	S	S	S	S	S	S	S	[39]
	351E	E	E	E	E	E	E	E	E	E	E	E	E	E	E	E	E	E	E	E	E	[40]
	356R	R	R	R	R	R	R	R	R	R	R	R	R	R	R	R	R	R	R	R	R	[41]
HA ^a^	165N	N	N	N	N	N	N	N	N	N	N	N	N	N	N	N	N	N	N	N	N	[25]
	174H	H	H	H	Q	H	H	H	H	H	H	H	H	H	H	H	H	H	H	H	H	[42]
	226L	L	L	L	L	L	L	L	L	L	L	L	L	L	L	L	L	L	L	L	L	[14]
	316S	S	S	S	S	S	S	S	S	S	S	S	S	S	S	S	S	S	S	S	S	[34]
NA	62-64	—	—	—	—	—	—	—	—	—	—	—	—	—	—	—	—	—	—	—	—	[34]
M1	30D	D	D	D	D	D	D	D	D	D	D	D	D	D	D	D	D	D	D	D	D	[43]
	215A	A	A	A	A	A	A	A	A	A	A	A	A	A	A	A	A	A	A	A	A	[43]
NS1	149A	A	A	A	A	A	A	A	A	A	A	A	A	A	A	A	A	A	A	A	A	[44]

^a^ H3 numbering. — represents deletion of amino acid.

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
