# Peer review of "A Well-Defined H9N2 Avian Influenza Virus Genotype with High Adaption in Mammals was Prevalent in Chinese Poultry Between 2016 to 2019"

_viruses, 2020, doi:10.3390/v12040432_

Round 1

Reviewer 1 Report

General Comments

The manuscript “A well-defined H9N2 avian influenza virus genotype with high adaption in mammals was prevalent in Chinese poultry between 2016 to 2019” describes a thorough investigation of recent H9N2 influenza A viruses derived from poultry in China with respect to replication and mutations in mice as a model for mammalian adaptation. The methods, results, and interpretation are well done. The only major concern I have is that the methods need to be more clearly stated with respect to the overall experimental design. I discuss that broadly below and then provide specific comments. I’ve also included minor suggestions to potentially improve some of the English.

The design of the study is not clear and is described inconsistently. The authors should consider adding a subsection addressing the experimental design after the “Viruses, cells, and animals” subsection. How many isolates were tested? How many dilutions were tested (were these 1:10 dilutions?)? How many MDCK or egg replicates were tested for each virus/dilution? For mice, what is a group (I assume groups are associated with the different viral strains, but this should be stated explicitly)? How many animals were tested per group? It seems like it was 11 total mice (3 euthanized on day 3, 3 euthanized on day 6, and 5 monitored for weight for 2 week. But, the figure legend for Fig. 2 indicates that 3 mice each were euthanized on days 3, 5, and 7.

Specific Comments

Line 75: Recommend adding a citation for Reed and Muench.

Line 88: What does it mean that the experiment was repeated at least twice? Does this mean that 2 groups of mice were tested for each virus? Were more groups tested for some of the strains?

Line 91: Was the amount/weight of lung tissue tested standardized?

Line 106: Do not need to repeat that MEGALIGN was used for analysis (also indicated in Line 102).

Line 108: The mouse data include standard deviations, but the egg and MDCK data do not. It is not clear where the results of these statistical analyses are reported in the Results section.

Line 118: Recommend clarifying that Sixteen isolates grew to >= 10^4 in MDCK cells.

Line 126. This sentence is confusing. Consider changing to something like, Lung virus titers were determined for three mice each on 3 and 6 dpi for each group as summarized in Figure 3.

Line 133: The figure legend should be carefully reviewed for accuracy.

Line 206: It would be useful here for the authors to specify the analysis that was used to develop Figure 6. Also, the authors might consider a different color scheme to aid readers with red-green color blindness. Nice figure!

Line 213: Please include a description of M, I V, P, N, L and A in the figure legend (or clarify in Line 210).

Minor suggestions for consideration

Line 18: Recommend replacing “adaption” with the much more common “adaptation” throughout the manuscript.

Line 21: Recommend replacing “lesion of lungs” with “lung lesions.”

Line 21: Recommend deleting “viral genome of the” since assessing viral replication is an assessments of infectious particles, not their genomes.

Line 25: Recommend deleting “Further” as no genomic analysis has been reported up to this point.

Line 27: Recommend changing “Additional 9 amino acids residues” to “An addition nine amino acid residues.”

Line 31: Recommend changing “mutations that associated with host adaption should be strengthen” to “mutations associated with host adaptation should be strengthened.”

Line 37: Recommend deleting the word “have.”

Line 38: Recommend changing “become” to “have become.”

Line 52: Recommend changing “and perimental” to “and the surrounding.”

Line 53: Recommend changing “investigated” to “investigated.”

Line 54: Recommend deleting “of mutations” which is already implied by saying you’re investigating adaptation to mammals.

Line 61: Recommend adding a space after the first comma.

Line 73: Recommend changing “were” to “was.”

Line 74: Recommend changing “test” to “tested.”

Line 80: Recommend removing the space between 37 and the degree symbol.

Line 87: Write out acronyms at first usage, e.g., post inoculation (p.i.). Also, both p.i. and dpi are used in the manuscript. Authors should consider standardizing to one or the other throughout the manuscript.

Line 101: Recommend changing “according to the” to “according to a.”

Line 114: Change “vial” to “viral.”

Line 122: Change TICD to TCID; change “were” to “are.”

Line 144: Recommend changing the sentence for clarity from “Influence of the prevalent H9N2 AIV strains..” to “Influence of infection with the prevalent H9N2 strains…” Also, change “were” to “was.”

Line 151: Change “weight standard deviations” to “weight and standard deviations.”

Line 152: Use subscripting and superscripting.

Line 164: Recommend changing “slightly thicken” to “slight thickening” and change “were” to “was.”

Line 165: Change “thicken” to “thickening” and “were” to “was.”

Line 172. Here the authors use H&E, but HE was used in the text. Please standardize.

Line 174: Change “thicken” to “thickening.”

Line 175: Recommend changing “cell which exhibits a small mount of effusion or point infiltration and even forms inflammatory cuffs.” to “cells which exhibit a small amount of effusion or point infiltration and even form inflammatory cuffs.”

Line 176 and 177: Recommend changing “indicate” to “indicates.”

Line 189: Recommend changing “link” to “links.”

Line 208: Recommend changing “maybe” to “may be.”

Line 220: Recommend changing “use” to “used” and deleting the second “the.”

Line 239: Recommend changing “lung” to “lungs.”

Line 244: Recommend changing “group” to “groups” and deleting the “in.”

Line 245: Recommend changing “evolved” to “evolve.”

Line 246: Recommend changing “strengthen” to “strengthened.”

Line 249: Recommend changing “typically” to “typical.”

Line 249: Recommend changing “typically” to “typical.”

Line 278: Recommend deleting the “that.”

Line 283: Recommend changing “lung” to “lungs.”

Line 287: Recommend changing “Additional” to “Additionally.”

Line 289: Recommend changing “on” to “of”

Line 290: Recommend changing “newly identified mutations that associated with host adaption should be strengthen.” to “newly identified mutations that are associated with host adaptation should be done.”

Author Response

C: Comment; R: Response

General Comments

The manuscript “A well-defined H9N2 avian influenza virus genotype with high adaption in mammals was prevalent in Chinese poultry between 2016 to 2019” describes a thorough investigation of recent H9N2 influenza A viruses derived from poultry in China with respect to replication and mutations in mice as a model for mammalian adaptation. The methods, results, and interpretation are well done. The only major concern I have is that the methods need to be more clearly stated with respect to the overall experimental design. I discuss that broadly below and then provide specific comments. I’ve also included minor suggestions to potentially improve some of the English.

The design of the study is not clear and is described inconsistently. The authors should consider adding a subsection addressing the experimental design after the “Viruses, cells, and animals” subsection. How many isolates were tested? How many dilutions were tested (were these 1:10 dilutions?)? How many MDCK or egg replicates were tested for each virus/dilution? For mice, what is a group (I assume groups are associated with the different viral strains, but this should be stated explicitly)? How many animals were tested per group? It seems like it was 11 total mice (3 euthanized on day 3, 3 euthanized on day 6, and 5 monitored for weight for 2 week. But, the figure legend for Fig. 2 indicates that 3 mice each were euthanized on days 3, 5, and 7.

R: We have added a subsection addressing the experimental design after the “Viruses, cells, and animals” subsection. Number of virus isolates was added in line 80. Detail method for MDCK, egg, or mice experiments were also added in the subsection accordingly. A total of 11 mice were used in each group: 3 euthanized on day 3, 3 euthanized on day 6, and 5 monitored for weight for 2 week. Figure legend for Fig. 2 was revised according to the experimental design.

Specific Comments

C1. Line 75: Recommend adding a citation for Reed and Muench.

A1. Citation for Reed and Muench has been added in line 363. The reference was cited in line 96 and line 104.

C2. Line 88: What does it mean that the experiment was repeated at least twice? Does this mean that 2 groups of mice were tested for each virus? Were more groups tested for some of the strains?

A2. The animal experiments were repeat for twice, and the experiment was repeated one more time if the virus was not detected in the undiluted lung samples of the inoculated mice of both repeats.

C3. Line 91: Was the amount/weight of lung tissue tested standardized?

A3. Half of the lungs from each mice were collected and tested for virus titers.

C4. Line 106: Do not need to repeat that MEGALIGN was used for analysis (also indicated in Line 102).

A4. MEGALIGN has been deleted.

C5. Line 108: The mouse data include standard deviations, but the egg and MDCK data do not. It is not clear where the results of these statistical analyses are reported in the Results section.

A5. The egg and MDCK data were used to preliminary evaluate the viral replication capacities and titers, and they have not been statistical analyzed between groups. BALB/c mice were employed as a model to evaluate the replication capacity and potential pathogenicity in the mammalian hosts of the prevalent H9N2 AIV isolates

C6. Line 118: Recommend clarifying that Sixteen isolates grew to >= 10^4 in MDCK cells.

A6. We have revised the description in line 151.

C7. Line 126. This sentence is confusing. Consider changing to something like, Lung virus titers were determined for three mice each on 3 and 6 dpi for each group as summarized in Figure 3.

A7. We have revised the description in line 156.

C8. Line 133: The figure legend should be carefully reviewed for accuracy.

A8. Figure legends have been revised according to the reviewer’s suggestions.

C9. Line 206: It would be useful here for the authors to specify the analysis that was used to develop Figure 6. Also, the authors might consider a different color scheme to aid readers with red-green color blindness. Nice figure!

A9. Figure 6 was developed based on the viral titers in mice lungs and comparison of deduced amino acid sequences from eight segments of H9N2 AIV. Thanks for the reviewer’s suggestion. We have revised the figure with a different color scheme to aid readers with red-green color blindness.

C10. Line 213: Please include a description of M, I V, P, N, L and A in the figure legend (or clarify in Line 210).

A10. A description of M, I V, P, N, D, L and A were added in the figure legends in line 484.

Minor suggestions for consideration

C1. Line 18: Recommend replacing “adaption” with the much more common “adaptation” throughout the manuscript.

A1. “adaptation” has been replaced by “adaption” throughout the revised manuscript.

C2. Line 21: Recommend replacing “lesion of lungs” with “lung lesions.”

A2. “lesion of lungs” has been replaced by “lung lesions” in line 21.

C3. Line 21: Recommend deleting “viral genome of the” since assessing viral replication is an assessments of infectious particles, not their genomes.

A3. We have revised the sentence in line 21 to avoid ambiguity. The whole genomic sequence of the viruses were analyzed here.

C4. Line 25: Recommend deleting “Further” as no genomic analysis has been reported up to this point.

A4. As C3 has been said, the whole genomic sequence of the viruses were analyzed in line 21.

C5. Line 27: Recommend changing “Additional 9 amino acids residues” to “An addition nine amino acid residues.”

A5. “Additional 9 amino acids residues” has been replaced by “An addition nine amino acid residues” in line 29.

C6. Line 31: Recommend changing “mutations that associated with host adaption should be strengthen” to “mutations associated with host adaptation should be strengthened.”

A6. The sentence has been corrected according to the reviewer’s suggestion in line 34.

C7. Line 37: Recommend deleting the word “have.”

A7. We deleted the word “have.”

C8. Line 38: Recommend changing “become” to “have become.”

A8. “become” has been replaced by “have become” in line 43.

C9. Line 52: Recommend changing “and perimental” to “and the surrounding.”

A9. “and perimental” has been replaced by “and the surrounding” in line 60.

C10. Line 53: Recommend changing “investigated” to “investigated.”

A10. “investigated” has been replaced by “investigate” in line 61.

C11. Line 54: Recommend deleting “of mutations” which is already implied by saying you’re investigating adaptation to mammals.

A11. We deleted “of mutations”.

C12. Line 61: Recommend adding a space after the first comma.

A12. A space has been added after the first comma in line 71.

C13. Line 73: Recommend changing “were” to “was.”

A13. “were” has been replaced by “was” in line 93.

C14. Line 74: Recommend changing “test” to “tested.”

A14. “test” has been replaced by “tested” in line 94.

C15. Line 80: Recommend removing the space between 37 and the degree symbol.

A15. The space between 37 and the degree symbol has been removed in line 102.

C16. Line 87: Write out acronyms at first usage, e.g., post inoculation (p.i.). Also, both p.i. and dpi are used in the manuscript. Authors should consider standardizing to one or the other throughout the manuscript.

A16. The acronyms of “dpi” has been written out at first usage in line 111. “dpi” has been used throughout the manuscript.

C17. Line 101: Recommend changing “according to the” to “according to a.”

A17. “according to the” has been replaced by “according to a” in line 128.

C18. Line 114: Change “vial” to “viral.”

A18. “vial” has been replaced by “viral” in line 148.

C19. Line 122: Change TICD to TCID; change “were” to “are.”

A19. The words have been corrected accordingly.

C20. Line 144: Recommend changing the sentence for clarity from “Influence of the prevalent H9N2 AIV strains..” to “Influence of infection with the prevalent H9N2 strains…” Also, change “were” to “was.”

A20. The sentence has been corrected according to the reviewer’s suggestion. “were” has been replaced by “was” in line 164.

C21. Line 151: Change “weight standard deviations” to “weight and standard deviations.”

A21. “weight standard deviations” has been replaced by “weight and standard deviations” in line 458.

C22. Line 152: Use subscripting and superscripting.

A22. Subscripting and superscripting were used for log10EID50 and TCID50.

C23. Line 164: Recommend changing “slightly thicken” to “slight thickening” and change “were” to “was.”

A23. The words have been corrected accordingly in line 180.

C24. Line 165: Change “thicken” to “thickening” and “were” to “was.”

A24. The words have been corrected accordingly in line 181 and line 182.

C25. Line 172. Here the authors use H&E, but HE was used in the text. Please standardize.

A25. H&E was used throughout the manuscript.

C26. Line 174: Change “thicken” to “thickening.”

A26. “thicken” has been replaced by “thickening” in line 466.

C27. Line 175: Recommend changing “cell which exhibits a small mount of effusion or point infiltration and even forms inflammatory cuffs.” to “cells which exhibit a small amount of effusion or point infiltration and even form inflammatory cuffs.”

A27. The sentence has been corrected accordingly in line 468.

C28. Line 176 and 177: Recommend changing “indicate” to “indicates.”

A28. “indicate” has been replaced by “indicates” in line 469 and 470.

C29. Line 189: Recommend changing “link” to “links.”

A29. “link” has been replaced by “links” in line 199.

C30. Line 208: Recommend changing “maybe” to “may be.”

A30. “maybe” has been replaced by “may be” in line 213.

C31. Line 220: Recommend changing “use” to “used” and deleting the second “the.”

A31. The words have been corrected accordingly in line 220.

C32. Line 239: Recommend changing “lung” to “lungs.”

A32. “lung” has been replaced by “lungs” in line 243.

C33. Line 244: Recommend changing “group” to “groups” and deleting the “in.”

A33. The words have been corrected accordingly in line 249.

C34. Line 245: Recommend changing “evolved” to “evolve.”

A34. “evolved” has been replaced by “evolve” in line 250.

C35. Line 246: Recommend changing “strengthen” to “strengthened.”

A35. “strengthen” has been replaced by “strengthened” in line 252.

C36. Line 249: Recommend changing “typically” to “typical.”

A36. “typically” has been replaced by “typical” in line 255.

C37. Line 278: Recommend deleting the “that.”

A37. We have deleted the word “that”.

C38. Line 283: Recommend changing “lung” to “lungs.”

A38. “lung” has been replaced by “lungs” in line 295.

C39. Line 287: Recommend changing “Additional” to “Additionally.”

A39. “Additional” has been replaced by “Additionally” in line 300.

C40. Line 289: Recommend changing “on” to “of”.

A40. “on” has been replaced by “of” in line 302.

C41. Line 290: Recommend changing “newly identified mutations that associated with host adaption should be strengthen.” to “newly identified mutations that are associated with host adaptation should be done.”

A41. The sentence has been corrected according to the reviewer’s suggestion in line 304.

Reviewer 2 Report

In the present research article, entitled “A well-defined H9N2 avian influenza virus genotype with high adaption in mammals was prevalent in Chinese poultry between 2016 to 2019” authors have identified a well-defined H9N2 AIV genotype to have high adaptability in mammals.

For the purpose, they analyzed viral genotype based on phylogenetic analysis using the MEGALIGN program with the clustal alignment algorithm and the neighbor-joining method.

  1. Genetic and phylogenetic analysis method must be explained briefly.
  2. Referencing tool must be used to avoid double numbering.

Author Response

C: Comment; R: Response

Comments and Suggestions for Authors

In the present research article, entitled “A well-defined H9N2 avian influenza virus genotype with high adaption in mammals was prevalent in Chinese poultry between 2016 to 2019” authors have identified a well-defined H9N2 AIV genotype to have high adaptability in mammals.

For the purpose, they analyzed viral genotype based on phylogenetic analysis using the MEGALIGN program with the clustal alignment algorithm and the neighbor-joining method.

C1. Genetic and phylogenetic analysis method must be explained briefly.

A1. Genetic and phylogenetic analysis method has been explained briefly in line 128-136.

C2. Referencing tool must be used to avoid double numbering.

A2. References have been carefully checked to avoid double numbering.